# Iron and Targeted Iron Therapy in Alzheimer’s Disease

**DOI:** 10.3390/ijms242216353

**Published:** 2023-11-15

**Authors:** Jian Wang, Jiaying Fu, Yuanxin Zhao, Qingqing Liu, Xiaoyu Yan, Jing Su

**Affiliations:** Key Laboratory of Pathobiology, Department of Pathophysiology, Ministry of Education, College of Basic Medical Sciences, Jilin University, 126 Xinmin Street, Changchun 130012, China; wjian21@mails.jlu.edu.cn (J.W.); fujy21@mails.jlu.edu.cn (J.F.); yuanxinz22@mails.jlu.edu.cn (Y.Z.); liuqq22@mails.jlu.edu.cn (Q.L.); yanxy@jlu.edu.cn (X.Y.)

**Keywords:** Alzheimer’s disease, iron, hyperphosphorylated tau, β-amyloid plaque, ferroptosis, insulin resistance, iron chelators

## Abstract

Alzheimer’s disease (AD) is the most common neurodegenerative disease worldwide. β-amyloid plaque (Aβ) deposition and hyperphosphorylated tau, as well as dysregulated energy metabolism in the brain, are key factors in the progression of AD. Many studies have observed abnormal iron accumulation in different regions of the AD brain, which is closely correlated with the clinical symptoms of AD; therefore, understanding the role of brain iron accumulation in the major pathological aspects of AD is critical for its treatment. This review discusses the main mechanisms and recent advances in the involvement of iron in the above pathological processes, including in iron-induced oxidative stress-dependent and non-dependent directions, summarizes the hypothesis that the iron-induced dysregulation of energy metabolism may be an initiating factor for AD, based on the available evidence, and further discusses the therapeutic perspectives of targeting iron.

## 1. Introduction

Alzheimer’s disease (AD) is one of the most common neurodegenerative diseases worldwide, with 100 million cases expected to occur by 2050 [1]. The socioeconomic burden of AD is alarming and increasing, as there is no effective treatment for this disease [2]. AD is classified as familial or sporadic, with familial early onset AD accounting for 1–5% of cases and mutations in the presenilin 1 (*PSEN1*), presenilin 2 (*PSEN2*), and amyloid precursor protein (*APP*) genes. Sporadic late-onset AD accounts for 95% of all AD cases and occurs in patients aged >65 [3]. There were more than 300 mutations in *PSEN1* (221 pathogenic) and 80 mutations in *PSEN2* (19 pathogenic) [4]. Mutations in *PSEN1* lead to the most severe form of AD, with complete epimutations. Meanwhile, missense mutations in *PSEN2* can manifest as incomplete episodic mutations, and those with mutations in *PSEN2* have a relatively late age of onset compared to those with mutations in *PSEN1* [5]. Around 73 mutations (32 pathogenic) have been identified in the *APP* gene, and mutations result in an increased Aβ42/Aβ40 ratio and increased levels of total and phosphorylated tau protein in neurons [6]. It is worth noting that the most important gene associated with sporadic Alzheimer’s disease (sAD) is *APOE*, and alleles of *APOE* that are translated into protein isoforms increase the risk of developing Alzheimer’s disease [7]. However, not all individuals who carry this allele develop the disease; it is only a predisposing risk factor [8]. More than 40 risk alleles associated with Alzheimer’s disease have now been identified in genome-wide association studies (GWAS). These studies have helped elucidate the pathology associated with the relatively high risk of developing AD and provided significant insights into the pathogenesis of AD [9]. However, owing to the multiple and complex mechanisms involved in the development of AD, the initiating cause of numerous uncertain downstream events remains unknown.

Iron is absent in the brain at birth [10]; it increases rapidly between adolescence and middle age, and remains relatively stable thereafter [11]. Iron accumulation primarily occurs in the basal ganglia and other brain regions associated with motor function. During adulthood, the red nucleus, substantia nigra, and nucleus accumbens accumulate rapidly [12,13]. Whole-brain MRI studies have shown that in addition to deep gray matter, the precentral cortex, prefrontal cortex, and occipital cortex, which are involved in motor, cognitive, and visual functions, also accumulate iron with age [12,14]. Recent studies have shown that iron is deposited in the brains of patients with AD [15] and appears to underlie the pathological progression of AD [16]. The role of iron in this process is thought to be primarily due to its potent ability to induce oxidative stress [17]. Because iron (mainly Fe^2+^) has redox activity, it can form more damaging free radicals when intracellular iron accumulation occurs and can catalyze the decomposition of H_2_O_2_ or lipid peroxides through the Fenton and Haber–Weiss chemical reactions, respectively [18,19]. Importantly, evidence has shown that iron may also be involved in shaping the major pathologies of AD in a non-oxidative stress-dependent manner. Therefore, exploring the role of iron in the initiation of sAD pathogenesis is of equal interest.

In this review, we aim to provide insights into the role of iron in Aβ pathology, tau phosphorylation, ferroptosis, and imbalances in the brain energy metabolism. It was concluded that the dysregulation of the energy metabolism involving iron is more likely to be a major initiating factor in the development of AD. The remainder of this section discusses research advances in the use of iron as a therapeutic target for AD.

## 2. Physiological Iron Transport

The process of iron transport in the cells has been studied in detail. Iron transport across the luminal membrane of the capillary endothelium is transferrin/transferrin receptor 1 (Tf/TfR1) dependent. Two Fe^3+^ atoms in the blood bind to Tf before binding to TfR1, which then enters the cytosol via lattice protein-mediated endocytosis of the Tf-TfR1 complex [20]. The interaction between Fe^3+^ and Tf is pH-dependent, with proton pumps inducing endosome acidification to a pH of 5.5, which triggers the dissociation of Fe^3+^ from Tf [21]. The iron reductase six-transmembrane epithelial antigen of prostate 3 (STEAP3) reduces Fe^3+^ in the endosome to Fe^2+^, which is then transported out of the endosome by divalent metal transporter 1 (DMT1) [22]. Excess iron is stored in the cytosolic ferritin (FT) or labile iron pool (LIP) [23]. FT is composed of 24 subunits of heavy chain (FTH) ferritin and light chain (FTL) ferritin, both of which play important roles in maintaining iron homeostasis [24]. FTH is more active in iron turnover and is present in tissues with high iron oxidation activity. It catalyzes the oxidation of Fe^2+^ to Fe^3+^ without the formation of excessive oxygen radicals, decreases Fe^2+^ accumulation, and is considered a major cytoprotective agent [25]. For example, more FTH is present in the brain to prevent potential free Fe^2+^ damage [26]. In contrast, FTL is associated with long-term iron storage and present in tissues with storage functions and low iron oxidation activity (e.g., the spleen, liver, and placenta) [27]. After the release of Fe^3+^, non-iron-bound Tf (apo-Tf) returns to the luminal membrane alongside TfR1. Subsequently, apo-Tf is released into the blood, thus ensuring efficient cellular iron transport [28]. Excess intracellular iron is then released into the extracellular space by ferroportin1 (Fpn1) and taken up by the apo-Tf present in other cells of the brain [29]. Non-transferrin-bound iron (NTBI) represents another physiological form of iron. In many cases, NTBI can be chelated by molecules such as citrate and then taken up by astrocytes, which are closely associated with brain capillary endothelial cells [30].

## 3. Mitochondria: Intracellular Iron Stores

Mitochondria are a major hub for iron metabolism, utilization, and storage and are a major site of oxidative stress. Mitochondria are the sole sites of heme synthesis, and iron-containing heme is an essential component of hemoglobin and an important cofactor involved in the electron transport chain [31]. Mitochondria are also the main sites of Fe-S cluster synthesis, which is essential for electron transport in the oxidative phosphorylation process. Thus, mitochondria require continuous iron uptake to maintain heme and Fe-S cluster synthesis while avoiding high levels of reactive oxygen species (ROS) production.

Mitoferrin 1/2 (Mfrn 1/2) is an important protein that regulates iron entry into the mitochondria [32]. Christenson et al. purified recombinant Mfrn1 in vitro and demonstrated that Mfrn1 transports free iron, but not chelated iron complexes, into the mitochondria [33]. In an AD model of the nematode *Hidradenitis elegans*, the knockdown of *MFN1* reduced mitochondrial iron content and mitochondrial ROS levels, which slowed the progression of AD [34]. In addition, Mfrn1 is required for brain energy metabolism and hippocampus-dependent memory and cannot be completely replaced by Mfrn2 or other unknown iron ion carriers [35]. This suggests that Mfrn1 plays an important role in mitochondrial iron metabolism. In addition, the “kiss and run” hypothesis (transient contact between endosomes containing iron-bound Tf and mitochondria) may be a potential mechanism for iron entry into mitochondria. This hypothesis was previously restricted to reticulocytes [36,37]. However, Das et al. observed direct nanometer-resolution interactions between Tf endosomes and non-erythroid mitochondria, suggesting that the “kiss and run” process may be widespread in a variety of cells [38] (Figure 1). Nevertheless, whether this pathway plays a role in brain cells such as neurons remains unclear.

Mitochondrial iron is mainly stored as mitochondrial ferritin (FtMt) and is highly expressed in cells characterized by high energy consumption, such as neurons [39]. The human FtMt sequence shares 79% homology with FTH, but unlike cytosolic FT, FtMt mRNA is deficient in iron response elements (IREs) [40]. In rat hippocampal neurons, mitochondria release more cytochrome C into the cytoplasm when FtMt expression is downregulated, causing mitochondria-dependent apoptosis [41,42]. Wang et al. found that in human IMR-32 neuroblastoma cells, the increase in FtMt expression was significantly accelerated when the cells were treated with the combination of H_2_O_2_ and Aβ compared to H_2_O_2_ alone. Furthermore, the overexpression of FtMt in IMR-32 cells prevented H_2_O_2_-induced cell death [43]. Further studies revealed that a high expression of FtMt controlled ROS generation by regulating mitochondrial iron availability. Additionally, it was found that FtMt attenuated Aβ-induced oxidative damage by redistributing iron from the cytosol to the mitochondria, leading to a reduction in cytoplasmic iron levels, suggesting that FtMt plays a protective role in cells characterized by iron homeostasis and respiratory defects [41,42]. However, Lu et al. found that MtFt induces increased cellular ROS production and cellular damage following treatment with tert-butyl hydrogen peroxide (tBHP) treatment [44].

Mechanistically, tBHP leads to more persistent oxidative stress because H_2_O_2_ is metabolized more rapidly than organic hydrogen peroxide in cells [45]. This prolonged oxidative stress stimulates high MtFt expression, leading to a prolonged decrease in cytoplasmic ferritin content, a compensatory increase in TfR, and an increase in TF-TfR-mediated iron uptake. Consequently, this ultimately leads to an increase in total cellular iron levels, which in turn causes oxidative stress injury [44]. Thus, FtMt may respond to oxidative stress that occurs early by regulating the spatial distribution of iron in the cell. However, as the disease progresses and the cell experiences sustained oxidative stress induced by factors such as Aβ, the regulation of cytoplasmic iron levels by FtMt may exacerbate cellular iron accumulation by increasing the cellular uptake of external iron, ultimately leading to more severe oxidative stress injury.

## 4. Impairment of the Blood–Brain Barrier (BBB) Is an Important Prerequisite for Iron Accumulation in the Brain

The brain is protected by a specialized structure known as the BBB. It comprises a compact layer of endothelial cells surrounded by stalks of astrocytes forming the neurovascular unit, which primarily provides structural and functional support [46,47]. Physiologically, many of the proteins and enzymes in these cells play an important role in protecting the brain from harmful polar molecules circulating in the bloodstream, thereby maintaining the precise regulation of the microenvironment within the brain [48]. Using MRI, Damulina et al. found higher iron concentrations in the deep gray matter and neocortical areas of the brain in patients with AD than in healthy controls, and the accumulation of temporal lobe iron levels over time was positively associated with cognitive decline in patients with AD [49]. Elevated iron levels were observed in the basal ganglia, especially in the caudate nucleus, nucleus accumbens, and pallidum, in the brains of patients with AD [50,51]. However, it is unlikely that this elevated iron level is due to a systemic increase in iron levels, as the entry of circulating iron into the brain is tightly regulated by the BBB [52]. Thus, abnormal BBB permeability may be a key factor in brain iron accumulation.

Aging can lead to increased BBB permeability, which may contribute to higher levels of iron in the brain [53]. This may be a potential mechanism by which aging increases the risk of developing AD. Major pathological changes associated with Alzheimer’s disease also compromise the integrity of the BBB. Li et al. observed elevated BBB permeability and reduced pericyte numbers in APP/PS1 transgenic mice. It was found that CD36 (which promotes vascular amyloid deposition and leads to vascular brain injury) and Aβ co-localized. Additionally, it was observed that Aβ upregulated CD36 expression within pericytes in the BBB, which in turn led to BBB destruction through mitochondrial autophagy induced by mitochondrial damage [54].

Aβ uptake by BBB pericytes may be a protective mechanism for the brain in response to Aβ deposition; however, the protective mechanism of Aβ phagocytosis by pericytes inadvertently becomes an “accomplice” in the accumulation of harmful polar molecules and iron accumulation in the brain. In addition, in a diabetic mouse model, BBB dysfunction was found to precede cognitive decline and neurodegeneration in mice [55]. Therefore, BBB integrity is an important prerequisite for the development of neurodegeneration and cognitive impairment.

## 5. Iron and Aβ

According to the currently dominant amyloid cascade hypothesis, abnormal extracellular accumulation of Aβ in the brain may lead to its aggregation into insoluble β-sheet protein structures, and these oligomers reorganize as protofibrils in amyloid plaques [56]. Aβ is a 39–42 amino acid peptide whose precursor is derived from APP. APP is a highly conserved protein that promotes synapse formation, dendritic sprouting, and neuronal migration [57]. APP is normally cleaved by α-disintegrin and metalloproteinase, initiating a non-amyloidogenic pathway that forms the APP intracellular domain and soluble, extracellularly secreted APPsα fragments [58]. When APP is hydrolyzed by the beta-site amyloid precursor protein cleaving enzyme-1 (BACE-1) and γ-secretase complex, deleterious Aβ accumulation occurs as a result [59].

Recently, an η-secretase with unclear function was identified. The carboxy-terminal fragment produced by the cleavage of APP by η-secretase was enriched in dystrophic neurons in mouse models of AD and in the brain of human AD, which may be involved in neuronal damage processes [60,61].

### 5.1. Iron Promotes the Expression of the Aβ Precursor APP and the Abnormal Cleavage Process of APP

The mRNA of APP contains an IRE in its 5′-UTR, making its translation extremely dependent on the intracellular iron concentration. It exhibits a strong preference for iron over copper and remains unresponsive to zinc [62]. Iron chelators inhibit APP translation, while the influx of iron reverses this inhibition [63]. Meanwhile, Zheng et al. found that DMT1 silencing in human neuroblastoma cells reduced Fe^2+^ inward flow, which in turn led to reduced APP expression and Aβ secretion [64]. Thus, if Aβ pathology is indeed a central aspect of AD, these findings would support a role for increased iron levels in APP expression and AD pathogenesis. In turn, APP has a regulatory effect on cellular iron content.

Researchers found that APP has ferroxidase activity mediated by a conserved FTH-like active site and interacts with ferroportin to catalyze the oxidation of Fe^2+^ to Fe^3+^ and promote Fe^3+^ binding to FT [65]. APP knockdown in primary neurons significantly induces iron retention, whereas increased APP expression promotes iron export to the extracellular compartment [65]. From this perspective, an appropriate concentration of iron maintains the expression level of APP for its normal physiological function. APP also regulates intracellular iron content to maintain its stability by interacting with FPN, which leads to the accumulation of iron beyond the limit of cellular metabolism, resulting in increased APP expression. Furthermore, the excessive increase of iron not only increases the expression of the Aβ substrate APP but also increases the activity of γ-secretase [66], which promotes the production of Aβ.

As an important link in Aβ formation, human AD brain extracts have been found to contain high levels of BACE-1 activity. Interestingly, both intracellular and extracellular experiments conducted by Chen et al. showed that increased iron levels decreased BACE-1 activity in a dose-dependent manner [67]. However, Xiong et al. found that the simultaneous presence of Fe^3+^ and Aβ42 promoted the upregulation of BACE-1 in the retina [68]. Notably, in APP transgenic (APP-tg) mice and AD brains, high levels of BACE-1 were observed in neuroinflammatory dystrophic regions surrounding the core of Aβ42-positive plaques, and this protein was found to be co-localized with neuronal proteins. This suggests that Aβ induces BACE-1 expression in peripheral neurons and that elevated BACE-1 is likely triggered by the amyloid pathway and is incidental to advanced AD. This may explain the discrepancy between the studies of Chen and Xiong, in that iron accumulation does not act as an initiator of BACE-1 expression, but rather becomes a driver of high BACE-1 expression in peripheral nerve regions after the initial formation of Aβ, which in turn drives a positive feedback loop of Aβ expression.

In APP-tg mice, FTL immunoreactivity is initially distributed throughout the brain and accumulated in the core of amyloid plaques as the disease progresses [69]. This change in the spatial location of FTL expression in AD progression seems to corroborate that Fe^3+^ has an important role in the extensive Aβ formation phase. Thus, the presence of η-secretase as an alternative to BACE-1 in APP cleavage [61] may be upregulated when BACE-1 expression is inhibited by iron in the early stages of AD, thereby playing an important role in Aβ formation.

### 5.2. For Aβ to Exert Its Toxicity, Iron Is a Key Factor

In the preclinical phase of AD, patients develop elevated levels of 8-hydroxyguanine, a marker of nucleic acid oxidation. They also exhibit a compensatory increase in 8-oxoguanine glycosylase (a DNA damage repair enzyme) levels, although no obvious clinical manifestations of AD will have occurred at this stage [70]. Thus, oxidative stress may occur earlier than expected [71]. Notably, one study showed that elevated Aβ levels were associated with increased levels of oxidation products of proteins, lipids, and nucleic acids in the hippocampus and cortex of AD subjects [72]. However, brain regions with lower Aβ levels (such as the cerebellum) do not exhibit high levels of oxidative stress markers [73].

Using Fourier transform infrared microscopy, Benseny-Cases et al. found the co-localization of amyloid deposits and lipid peroxidation in brain tissue sections from patients with AD [74]. In samples from patients diagnosed with AD, plaques and their surroundings always showed the presence of oxidized lipids, while samples from individuals without AD showed lower levels of lipid oxidation than those from individuals with AD. Interestingly, in some non-AD individuals, plaques could be detected in the brain; however, their surrounding lipids demonstrated similar levels of oxidation as tissues without plaques [74]. This result suggests that the oxidative capacity of Aβ may play a more central role than fibrillar aggregation.

It has been shown that oxidative stress-related metal ions such as zinc, iron, and copper are present in Aβ, and when these redox-active metal ions bind to Aβ, rapid AD-related protein aggregation and toxic oligomer formation, as well as an excessive production of ROS, are observed [75,76,77]. Considering that iron is the most abundant metal in the brain [78], this oxidative capacity of Aβ may be mainly achieved by the redox activity of iron ions. Interestingly, Everett et al. found that Aβ was able to accumulate Fe^3+^ in amyloid aggregates and also reduce it to redox-active Fe^2+^ [79]. This Aβ-mediated shift in the iron redox state also explains, to some extent, the increased levels of oxidative stress characteristic of Aβ aggregation. Mechanistically, Everett et al. further showed that Aβ plaques are capable of converting ferrihydrite into redox-active substances rich in Fe^2+^ [80] and that the magnetite contained in the core of Aβ may be responsible for their significant catalytic properties [81]. Due to APP’s capacity to catalyze the oxidation of Fe^2+^ to Fe^3+^, the formation of Aβ via the sequential shearing of APP through BACE-1 and γ-secretase may hold significant implications, as it is possible that this aberrant APP modification imparts Aβ with the ability to alter iron redox activity. It has also been shown that a high expression of BACE-1 in AD reduces the activity of superoxide dismutase 1 in cells and promotes oxidative damage [82]. Thus, the dysregulation of the antioxidant system due to the aberrant cleavage of APP may be another pathway for increased oxidative stress.

## 6. Iron and Phosphorylation of Tau

Tau, a microtubule-associated protein, is another major player in AD. Tau is a hydrophobic protein that binds to microtubules and regulates neuronal microtubule stability and axonal transport [83]. Tau regulates its dissociation from microtubules through posttranslational modifications (PTMs) such as phosphorylation, truncation, acetylation, glycosylation, and ubiquitination at many different residues, thereby affecting neuronal function [84,85]. For example, in AD brains, the aberrant glycosylation of tau proteins may occur prior to phosphorylation and contribute to the hyperphosphorylation of tau proteins [86]. Tau truncation disrupts the “paper clip” structure of tau, increasing its tendency to form aggregates and promoting tau phosphorylation [87].

Tau441, the longest isoform of tau, possesses over 80 potential phosphorylation sites, making phosphorylation the most prevalent PTM of tau. These phosphorylation events primarily occur on serine, threonine, and tyrosine residues [88]. Depending on the type and location of amino acid residues, phosphorylation can have different physiological effects. For example, tau phosphorylation at Ser262, Thr231, and Ser235 inhibits its binding to microtubules by 35%, 25%, and 10%, respectively [89], whereas phosphorylation at Thr231, Ser396, and Ser422 promotes tau aggregation into filaments [90]. In AD, hyperphosphorylation at specific amino acid sites is common and may promote tau aggregation and disrupt synaptic function, leading to neuronal death and the propagation of tau pathology. Wallin et al. used several biophysical methods and found that unmodified full-length tau pathologically prevents Aβ40 aggregation and fibrosis in a subchemically stoichiometric, dose-dependent manner. Consequently, the decrease in unmodified tau caused by increased tau phosphorylation may be the cause of the increase in Aβ40 aggregation and protofibrosis [91].

### Iron Promotes Tau Phosphorylation through Multiple Pathways

Elevated iron levels have been observed in brain regions that accumulate neurofibrillary tangles (NFTs), such as the cortex and hippocampus, in patients with AD. Mechanistically, iron can generate tau oligomers through the formation of intermolecular coordination complexes mediated by phosphorylated amino acid residues [92,93]. Ahmadi et al. found, through electrochemical studies, that Fe^2+^ showed a more significant effect in inducing tau aggregation than Fe^3+^, and that Fe^2+^ also mediates tau interactions [94]. Fe^3+^ can induce the pathological enhancement of hyperphosphorylated tau oligomer function, allowing phosphorylated tau to bind to membrane lipids at nanomolar protein concentrations, exacerbating disease progression [95]. Thus, the presence of iron may have a greater impact on tau pathology than previously thought.

Although the intracellular antioxidant system can cope with redox-active Fe^2+^, the Fe^3+^ formed after Fe^2+^ is oxidized can continue to play a role in promoting tau neurotoxicity. Tau is also involved in APP-mediated iron transport. Tau transports APP to the cell membrane to stabilize the iron export channel, Fpn1 [96]. In in vitro experiments, the deletion of tau led to iron retention by reducing APP-mediated iron export [97]. Therefore, when large amounts of tau phosphorylation result in reduced normal tau levels, it may promote iron retention and exacerbate the progression of Aβ pathology by affecting the function of APP.

During NFT formation, iron accumulation-induced oxidative stress promotes tau phosphorylation. On the one hand, ROS generated by iron accumulation may lead to the formation of oligomeric tau by the binding of sulfhydryl-containing cysteines [97]. On the other hand, oxidative stress activates the additional phosphorylation of tau triggered by the PI3K-Akt-GSK-3β pathway [98] (glycogen synthase kinase 3β (GSK-3β), a member of the proline-directed protein kinase family, promotes tau phosphorylation). Interestingly, Apopa et al. applied confocal microscopic imaging analysis to show that iron nanoparticles increased the permeability of human microvascular endothelial cells and that the iron nanoparticle-induced production of ROS and microtubule remodeling were the main factors that enhanced permeability [99]. ROS induced the activation of Akt, which is one of the major kinases that inhibit GSK-3β [100]. Subsequently, Akt inhibited GSK-3β phosphorylation, thereby inhibiting the ability of GSK-3β to phosphorylate tau and leading to microtubule stabilization. Conversely, the inhibition of ROS reversed these effects [99]. This suggests that, at certain levels, ROS may play a protective role in inhibiting tau phosphorylation and aggregation; however, this role is rather limited. Overall, increased levels of ROS increase the permeability of vascular endothelial cells in the brain, allowing an increase in intracellular iron levels, which, in turn, further contributes to the pathological progression of AD.

## 7. Ferroptosis and AD

Ferroptosis is an iron-dependent, regulated form of cell death characterized by iron overload and lipid peroxidation as key metabolic features [101]. Lipid peroxidation is a highly reactive molecule that destroys cellular components, including lipids, proteins, and DNA, resulting in cell death [102]. Lipids, an important component of the brain, make up 40% to 75% of the brain’s dry weight [103]. Owing to the physiological functional needs of the brain, it is rich in unsaturated lipids and has a high demand for redox-active metals. Meanwhile, the elevated levels of free radicals in the brains of patients with AD create a favorable environment for lipid peroxidation. When the most toxic oxygen radical, ·OH, is produced in large quantities and the antioxidant system is dysregulated, ·OH binds to cell membranes or mitochondrial membranes containing polyunsaturated fatty acids (PUFAs). This interaction results in lipid peroxidation of the membranes, which in turn triggers ferroptosis, another mechanism by which neurodegeneration occurs in AD [104,105].

Bao et al. showed that when the *Fpn1* gene is deleted in large neurons of the cortex and hippocampus of mice, a distinct ferroptosis signature is observed, ultimately leading to AD-like hippocampal atrophy and memory deficits [106]. It has also been shown that Aβ-induced oxidative stress can further trigger ferroptosis. Zhang et al. found that Aβ25-35 induced PC12 cells to exhibit increased ROS levels, decreased GPX activity, increased malondialdehyde levels, and mitochondrial depolarization, ultimately leading to ferroptosis [107]. In addition, Aβ-induced ferroptosis in neuronal cells occurs with the upregulation of acyl-CoA synthase long-chain family member 4 (ACSL4) [107]. ACSL4 has a clear preference for arachidonic acid (AA) as its substrate, and 4-hydroxynonenal (4-HNE) is the major metabolite of AA [108]. It has been shown that 4-HNE increases γ-secretase activity and promotes Aβ production through a mechanism of covalent modification of γ-secretase subunit nicastrin [109]. This shows that Aβ and the ferroptosis it induces can be mutually reinforcing to further exacerbate the damage to neurons.

Tau pathology has also been linked to ferroptosis, and Wang et al. found that ferroptosis promotes tau aggregation through GSK-3β activation and proteasome inhibition [110]. In addition, AMP-activated protein kinase (AMPK) is an important factor in the regulation of ferroptosis, and its activation inhibits ferroptosis [111]. Wang et al. demonstrated that the upregulation of AMPK inhibited GSK-3β activation, which attenuated tau hyperphosphorylation and ameliorated AD-induced memory impairment [112]. This finding demonstrates that the main pathological process of AD is closely related to ferroptosis. Furthermore, given that the Fenton reaction, mediated by the presence of large amounts of redox-active divalent iron ions, is an important component in the occurrence of ferroptosis, Aβ plaques with high iron content as well as phosphorylated tau aggregates may increase susceptibility to ferroptosis.

## 8. Iron and Dysregulated Energy Metabolism in the Brain

### 8.1. Elevated Iron Levels Induce Brain Insulin Resistance (IR)

The high energy demands of the brain and limited energy storage relative to energy demand make the brain highly dependent on the supply of glucose in the blood, and insulin is critical for glucose utilization in the brain [113]. In addition to regulating glucose metabolism, insulin is involved in the secretion of cognitive neurotransmitters such as acetylcholine, norepinephrine, and epinephrine, which profoundly affect synaptogenesis and synaptic plasticity in the central nervous system (CNS) [114]. Insulin, as a macropeptide hormone, cannot passively pass through the BBB and, therefore, enters the brain mainly through the cerebrospinal fluid through sites lacking an effective BBB, such as the hypothalamus, or is mediated by insulin receptors on the vascular endothelium [115,116]. IR is defined as the targeted tissue failing to respond normally to insulin [117]. IR is also present in the brain and is strongly associated with defective glucose utilization in the peripheral systems, including obesity, type II diabetes, normal aging, and dementia [118,119].

Iron accumulation can lead to the development of IR in the brain with concomitant cognitive decline [120]. Wan et al. found that ferrous (Fe^2+^) chloride led to a reduction in the phosphorylation levels of key components in the insulin signaling pathway, including insulin receptor β, insulin signaling substrate 1, and phosphatidylinositol 3-kinase p85α in primary cultured neurons. In vivo experiments also showed that iron accumulation induced a disruption of insulin signaling [121]. This may be the mechanism by which iron accumulation leads to impaired insulin signaling in the brain. Hao et al. found that significant cognitive dysfunction was observed in a rat model of type I diabetes established by intraperitoneal injection of STZ, accompanied by a significant increase in Fe^2+^ levels [122]. In AD, iron and impaired insulin signaling reinforce each other and contribute to cognitive decline.

### 8.2. Dysregulated Insulin Signaling Precedes and Contributes to Aβ and Tau Pathology

One study found that IR predicted brain Aβ deposition in late middle age and was an independent risk factor for Aβ accumulation in the brains of older adults without dementia [123,124]. The IRS-PI3K-AKT pathway is a major pathway for impaired insulin signaling and plays a role in insulin signaling abnormalities and IR [125]. Biliverdin reductase-A (BVR-A), a unique serine/threonine/tyrosine kinase, is an upstream regulator of the insulin signaling cascade that facilitates the Akt-mediated inhibition of GSK-3β [126]. Barone et al. found that oxidative stress induced impairments in BVR-A activity in human AD brains and that this impairment occurred before the pathological accumulation of Aβ and tau [127]. Notably, the deletion of BVR-A after oxidative stress impairs the neuroprotective effect of Akt in inhibiting GSK-3β signaling [126]. In contrast, the activation of GSK-3β will initiate Aβ and tau pathological processes. Furthermore, Aβ competitively inhibits the binding of insulin to receptors and subsequent receptor phosphorylation, which is an important factor contributing to synaptic and dendritic spine damage [128]. This demonstrates that impaired insulin signaling also intersects with Aβ pathways and tau pathology, is an important potential target for the prevention and treatment of AD, and precedes the onset of Aβ deposition and tau phosphorylation (Figure 2).

In addition to oxidative stress, an individual’s bioenergetic status at a fundamental level also profoundly affects protein aggregation. In one study, Patel et al. showed that ATP acts as a “biological hydrotrope”. The hydrophobic nucleotide fraction is associated with hydrophobic protein fragments, whereas the hydrophilic phosphate group maintains the complex in a soluble state. At physiological ATP concentrations, its hydrotropic properties prevent the self-aggregation of proteins. When the ATP concentration drops to moderate levels, this hydrotropic property diminishes, and oligomers are formed. When ATP levels decrease, fibers begin to aggregate [129,130]. Treatment with insulin also has the effect of reducing Aβ deposition and tau hyperphosphorylation in the brain [121,131]. Thus, reduced ATP levels in the brain due to impaired insulin signaling, which make it difficult to maintain Aβ and tau oligomerization, are also an important prerequisite for AD progression. This further supports an imbalance in energy metabolism, including IR, which may be the earliest pathogenic factor leading to AD development.

## 9. Iron Accumulation throughout AD: From IR-Induced Impairment of Energy Metabolism to Aβ Deposition and Tau Phosphorylation

Based on the available evidence, we propose a hypothesis regarding the development of AD. Factors such as aging or hyperglycemia may cause increased BBB permeability, and if this is coupled with the dysregulation of iron-transport-related proteins, a subsequent accumulation of iron in the brain occurs. Iron accumulation in the brain induces oxidative stress, which progressively worsens as iron levels increase. In this process, mitochondria reduce ROS production through their iron storage function during the early stages of iron accumulation; however, this protective effect is limited, and mitochondria eventually become a central site for ROS generation instead of progressively increasing iron levels.

Oxidative stress further disrupts the BBB’s integrity, exacerbates iron accumulation, and creates a vicious cycle. Dysregulated insulin signaling combined with iron accumulation directly initiates and drives the progression of pathologies such as Aβ and tau phosphorylation, leading to the aggregation of these toxic proteins, which accumulate in processes that reinforce each other and induce more severe oxidative damage and even ferroptosis, resulting in a cascade of increased damage in AD patients. It is essential to note that not all forms of iron contribute to the generation of ROS. Nevertheless, regardless of whether iron is present in a redox-active form or not, it inflicts severe damage on neurons in the brain. Therefore, iron is a key factor in AD and plays an important role in its etiology and pathogenesis [121]. As a result, the targeted treatment of iron accumulation in the AD brain may be a promising prospect.

## 10. Advances in Iron Chelators in AD Therapy

Given the evidence of pathological iron accumulation in AD, iron chelators that restore iron homeostasis are suitable therapeutic agents. Chelation is the process by which the ions/molecules of a ligand bind to a central metal atom or ion to form a ring or toroidal structure. Based on the nature of the bond between the ligand and covalent atoms, ligands can be classified into three types: unidentate (one donor atom), bidentate (two donor atoms), and polydentate (more than two donor atoms). Polydentate ligands form five- to six-membered ring complexes that are more stable than monodentate ligand–metal complexes [132]. Effective iron chelators should have a low molecular weight, high selectivity, and properties that allow them to cross physiological and membrane barriers to sites of iron ion concentration. The removal of iron from its biological ligands results in the formation of a harmless and non-toxic complex and facilitates its excretion from deposition sites without depleting other metal ions. This process should be carefully considered for the selective removal of iron from specific regions of the brain, without causing systemic iron deficiency [133].

Deferoxamine (DFO), deferiprone (DFP), and deferasirox (DFX) are iron chelators approved by the U.S. Food and Drug Administration for treating iron overload [134]. Recent studies have shown that these compounds can reduce iron accumulation in patients with AD. DFO, the first drug used to treat iron accumulation, is a small-molecule hexadentate iron chelator that binds to iron in a 1:1 ratio [135]. Owing to its short half-life, DFO must often be administered by injection, resulting in poor patient compliance [136,137]. In addition, the available evidence does not fully support the ability of DFO to cross the BBB, and high doses are required for neuroprotection, a factor that is linked to the occurrence of serious side effects [138,139]. To overcome these challenges, the use of intranasal DFO for iron chelation has been validated in several studies to reverse iron-induced memory deficits by inhibiting aberrant APP cleavage, Aβ aggregation, tau phosphorylation, and neuronal ferroptosis in AD mouse models [140,141,142,143]. This “shortcut” delivery method that bypasses the BBB opens the door to the clinical application of iron chelators with larger molecular weights.

DFP, the first oral iron chelator, is a bidentate ligand that binds iron in a 3:1 ratio and has a short half-life [144]. In comparison to DFO, DFP can significantly cross the BBB, a feature that gives DFP a substantial advantage in CNS administration [145]. Rao et al. used DFP to significantly reduce anxiety-like behaviors in mice, along with a decrease in brain iron and insoluble tau polymer levels [146]. In a different study, Chand et al. administered a tacrine (a palliative drug for anti-AD therapy)–DFP mixture to inhibit Aβ aggregation induced by multiple pathways, while also demonstrating good free radical scavenging capabilities and exhibiting neuroprotective effects [147]. Meanwhile, DFX has been introduced into clinical practice as a second oral iron chelator with a long half-life and a tridentate iron chelator that binds iron in a 2:1 ratio [148]. Banerjee et al. significantly blocked age-related iron accumulation and TfR1 and ferritin overexpression in the brain when DFX was administered daily to aged rats. DFX treatment also significantly reversed altered Aβ peptide metabolism in the aging brain and reversed oxidative stress and inflammatory activation in the brain [149]. In addition, Kwan et al. found that DFX may also inhibit tau aggregation by reducing the iron that aggregates tau or by directly binding tau [150]. These results strongly suggest that these iron chelators have significant potential for the treatment of CNS disorders.

In light of the evidence above, several new iron chelators have been developed. For instance, Feng et al. synthesized novel deferric amine compounds (DFAs) with tunable backbones and flexibilities. These compounds significantly ameliorated iron accumulation in various mouse models, including hemochromatosis, high-iron diet-induced iron accumulation, and iron dextran-stimulated iron accumulation. Moreover, these could inhibit iron-induced ferroptosis by modulating intracellular signals that drive lipid peroxidation [151]. However, whether or not novel iron chelators, including DFAs, have potential for CNS administration remains to be explored.

In addition to iron chelators, studies are being conducted to identify alternative therapeutic options. Ascorbic acid is another relevant compound that has been investigated in recent years. It offers the advantage of being directly involved in the regulation of redox reactions and interacting with iron to regulate several metabolic pathways [152]. However, vitamin C has a weak ability to chelate iron, and ascorbate forms a complex 3+ with iron, which is subsequently reduced to iron 2+. As a result, this may promote the production of free radicals. Therefore, this compound is mostly used as an adjuvant to DFO therapy [153]. Similarly, antioxidant drugs such as vitamin E [154] and α-lipoic acid [155] are thought to be involved in iron regulation and may be effective in the treatment of AD.

Iron has been shown to compete with calcium for entry into cultured neuronal cells in vitro via voltage-gated calcium channels [156]. In their study, Bostanci et al. blocked L-type calcium channels to protect hippocampal and substantia nigra neurons from iron neurotoxicity [157]. Therefore, calcium channel blockers may be another group of potential adjuncts to iron depletion. However, the limited number of relevant studies is insufficient to confirm the safety and efficacy of this approach, and further studies are urgently needed to validate the feasibility of this approach.

On a different note, chloroquine (CQ) has also been shown to be a modest chelator of iron. Grossi et al. found that CQ treatment had modest but significant effects on absolute and relative brain concentrations of copper, zinc, and iron. This led to reduced brain Aβ deposition and the prevention of memory impairment [158,159]. In contrast, other studies have focused on refining the administration route using nanoparticles (NPs). NP-mediated drug delivery offers unique advantages over free drug administration, such as increased drug concentration in diseased tissues through active targeting; reduced toxic side effects in normal tissues; improved solubility, pharmacokinetics, and pharmacodynamic profiles of the drugs; and improved drug stability by reducing its degradation in the systemic circulation [160]. The transportation of iron chelators from the blood to the brain can be enhanced by encapsulating them within NPs or by covalently attaching them to their surfaces, and their feasibility in the treatment of AD has been demonstrated [161,162]. This new approach to chelation not only provides an effective means for the treatment of AD but also provides new insights into the pathophysiological mechanisms of AD and may play a role in other iron-mediated neurodegenerative diseases [163].

The loading of iron chelators with NPs has some limitations, leading to limited clinical applications in the CNS. For example, none of the targets of the BBB that mediate the interactions of NPs, TfRs, LDL receptors, and lactoferrin receptors [164] are uniquely expressed in the BBB. Therefore, iron chelators can enter other tissues and organs in large quantities alongside NPs before reaching the brain. However, their effects remain unknown. Furthermore, it has been shown that the complement system is highly activated in the senile plaques in AD brains [165,166]; therefore, it is crucial to ensure that foreign substances in these brains do not cause additional complement activation. Thus, further studies are needed to demonstrate the efficacy and safety of iron chelator nanoparticle systems for CNS administration and to evaluate their toxicity in more detail (Figure 3).

## 11. Conclusions

AD is a devastating progressive neurodegenerative disease in which patients show signs of memory impairment and cognitive deterioration due to progressive neuronal loss. Its pathogenesis is complex and involves multiple interacting factors. As mentioned earlier, it is not only oxidative stress but also the presence of excess iron in any form that increases the risk of cognitive dysfunction and memory deficits in AD. Therefore, the importance of iron and mitochondria in the pathology of AD requires further attention.

## Figures and Tables

**Figure 1 ijms-24-16353-f001:**
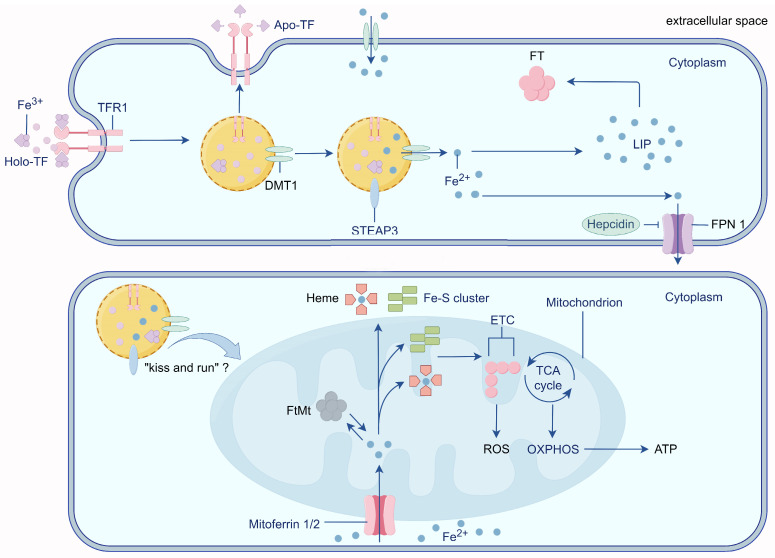
Mechanisms of cellular and mitochondrial iron transport. Extracellular Fe^3+^ enters the cell via Tf/TfR1 and is reduced to Fe^2+^ by STEAP3 in endosomes and stored in LIP and FT, and excess iron is released to the outside of the cell via FPN1. Mitochondria are important organelles in the regulation of iron homeostasis, and Fe^2+^ enters mitochondria via Mfrn1/2 to participate in the synthesis of iron-sulfur clusters and heme, which affects the energy status of mitochondria, and the “kiss and run” pathway may be another potential mechanism of iron entry into mitochondria. By Figdraw (https://www.figdraw.com).

**Figure 2 ijms-24-16353-f002:**
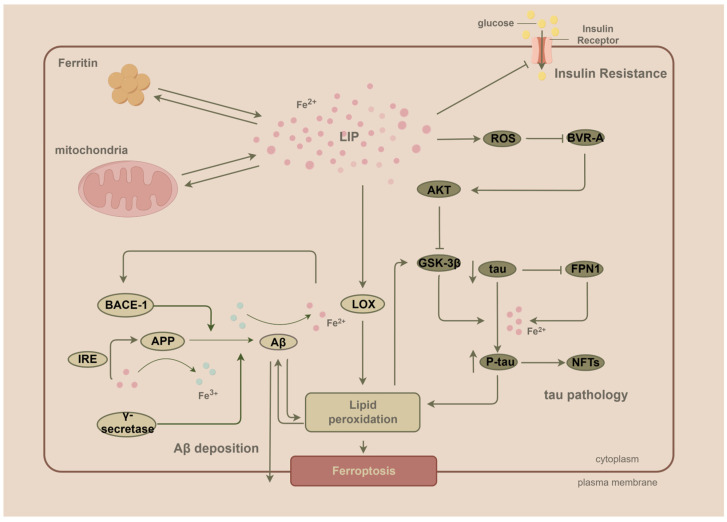
Iron is involved in the formation of the main pathological mechanisms of AD through a variety of pathways. Iron induces the development of IR and causes impaired cellular energy utilization and oxidative stress; iron increases the extracellular deposition of Aβ and lipid peroxidation levels by promoting APP expression and aberrant cleavage processes; iron promotes the hyperphosphorylation and aggregation of tua, the formation of NFTs, and the promotion of lipid peroxidation; and the above, combined with the accumulation of iron that occurs in the cell, ultimately causes ferroptosis in neurons. By Figdraw (https://www.figdraw.com).

**Figure 3 ijms-24-16353-f003:**
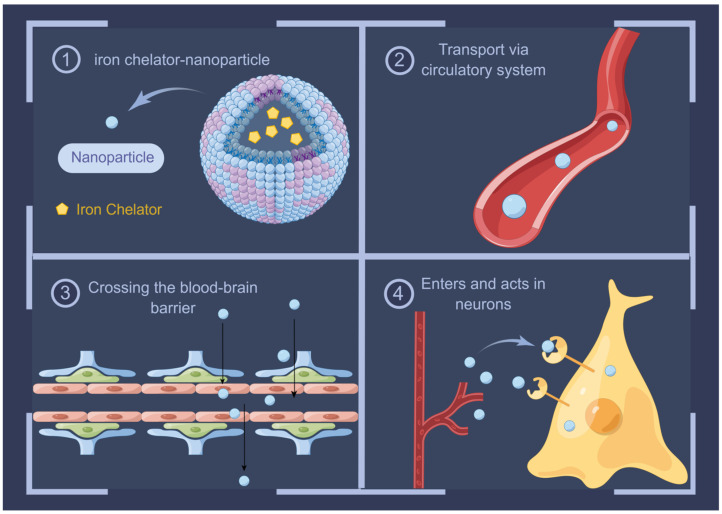
Encapsulation of iron chelators in appropriate NPs, relying on the BBB permeability possessed by NPs, will help to achieve targeted treatment of CNS iron accumulation. By Figdraw (https://www.figdraw.com).

## Data Availability

Not applicable.

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
