# Peer review of "Iron and Targeted Iron Therapy in Alzheimer’s Disease"

_ijms, 2023, doi:10.3390/ijms242216353_

Round 1

Reviewer 1 Report

Comments and Suggestions for Authors

This review highlights the importance of iron in AD progression. I have some suggestions to improve the manuscript.

Line 27-32, “Although AD is familial, familial AD accounts for less than 1% of all cases of AD and is associated with mutations in genes for key proteins and enzymes related to amyloid precursor protein (APP) metabolism (e.g., progerin) [4]. Unlike familial AD, the initiation of sporadic AD (sAD), which is the main form of AD present, remains hidden in numerous uncertain downstream events.

These two sentences are not well written, and confusing. In fact the whole paragraph on the general description of AD is far too short; it should be extended considerably. Please state that there are two main forms of AD, the familial early onset AD and the sporadic late onset AD. Furthermore, the mutations of APP and PSEN, rather than the progerin, should be mentioned. What is the role of MAPT and its PTMs on AD pathology? The recent GWAS studies that reveal the risk loci should also be discussed.

The authors should carefully proof read the manuscript; throughout the text there are sentences that are difficult to understand. For example, Line 226-227, “however, it was still possible in some non-AD individuals to plaques can still be detected in the brain of some non-AD individuals,”.

The inclusion of additional figures that illustrate the biological processes affected by iron will be helpful, in particular, the sections concerning the “Iron and phosphorylation of Tau” and the “Iron and dysregulated energy metabolism”.

Comments on the Quality of English Language

Moderate editing of English language required

Reviewer 2 Report

Comments and Suggestions for Authors

The work aims at analyzing the involvement of iron in the AD pathology with attention to the effects that are not directly linked to iron-induced oxidative stress. The approach may be interesting, but the mechanism by which iron affects APP, Tau and other is not discussed, and most of the described  actions rely on oxidative damage. In addition, there are many serious flaws in the work.

- Ferroptosis is considered to play a major role in the iron-dependent toxicity in AD, but it is only briefly indicated, without any detail on its mechanism. 

- Line 67.   “As in the brain, the proportion of FTH is increased [17]”. What does it mean?

- L. 69. “Iron uptake by endothelial cells is mediated by ferroportin1 (FPN1) across the basement membrane into the intracerebral circulation.”   Ferroportin is an iron exporter!!  The sentence is confusing, rephrase it.

- Mitochondria. Present understanding is the iron enters the mitochondria mainly through mitoferritin1 or 2.  The Kiss and run hypothesis is doubtful, and the idea that iron “infiltrates into the mitochondria” is original and but not supported by any data.  The whole paragraph should be updated and modified.

- The authors often refer to “brain iron overload”, which is incorrect!  Iron overload has not been observed in brain, but only in liver and other organs. More appropriate is “iron accumulation”.

- L. 239 “iron hydrides in FT” probably they mean ferrihydrite.

- L. 240 “the Fe3O4 crystals” probably they refer to magnetite.

- Paragraph 6.2 does not even mention iron and it is unclear how it fits in the work.

- Deferiprone (DFP) is the major iron chelator that has been shown to pass BBB and remove iron from the brain. It has been used in various clinical applications and trials. I am really surprised that it not even mentioned in this work. It must be included.

- The hypothesis to use nanoparticle to deliver iron chelators to the brain seems to have a low priority and practicality. Maybe a deeper discussion on the chemistry of iron chelators would be more appropriate. Figure 1 does not seem to improve present knowledge on the mechanism. More appropriate would be a fugure or table the illustrates the role of iron in AD. 

- The conclusions seem to focus on the mitochondria, but the text did not include data on iron removal from these organelles

Comments on the Quality of English Language

It is adequate

Reviewer 3 Report

Comments and Suggestions for Authors

In this review, Wang and collaborators correlated the role of iron with Alzheimer's disease. The review is very well-written and understandable. I am suggesting some point to improve this work.

The authors made a correlation between iron and AD, mainly based on the observation that iron accumulates in brain areas strongly associated with clinical symptoms of AD. Iron is a mediator for free radical reactions mediating the formation of lipid peroxides. Also, iron has been involved with cells sensitivity to ferroptosis, which is a type of cell death known to be dependent upon iron, which leads to the formation of specific lipid peroxides. It would be interesting if authors could describe the role of this type of cell death in AD. Also, it would be interesting to know if brain areas strongly associated with clinical symptoms of AD  where iron accumulates are more sensitibe to ferroptosis.

Minor corrections

Lines 38 and 39. There is a lack of connection between phrases. Please correct

Line 53. Please correct two Fe3+ by two Fe3+ atoms

Reviewer 4 Report

Comments and Suggestions for Authors

The manuscript ijms-2643445 entitled More than oxidative stress: the role of iron in Alzheimer's disease and targeted iron therapy by Jian Wang and coworkers is a review about the role of Iron in Alzheimer Disease. Alzheimer's disease (AD) remains the most common neurodegenerative disease worldwide. β-Amyloid plaque (Aβ) deposition and hyperphosphorylated tau and dysregulated energy metabolism in the brain are involved in pathogenesis and progression of AD.

So far, several evidences reported aberrant iron accumulation in different regions of the AD brain.The authors discuss the main mechanisms and recent advances of iron involvement in the above pathological processes, including iron-induced oxidative stress-dependent and non-dependent directions, summarize the hypothesis that dysregulation of energy metabolism involving iron may be an initiating factor of AD based on the available evidence, and also further discuss the prospects and obstacles of nanoparticles (NPs) combined with iron chelating agents for the treatment of AD.

The review work is well written, clear and informative.

However, one or two figures would be helpful.

One with the Iron metabolism in the neuronal cell and another with a schematic effect of iron on Ab, P-tau and metabolism evidencing the most important evidences.

Minor revision

Line 220: a space should introduced after the point.

Line 221: a space should introduced after the point.

Line 223-228: Interestingly, plaques and their surrounding tissue were always characterized by the presence of oxidized lipids in samples from patients diagnosed with AD; samples from non-AD individuals showed lower levels of lipid oxidation than those from AD individuals; however, it was still possible in some non-AD individuals to plaques can still be detected in the brain of some non-AD individuals, but plaques and their surrounding lipids show similar levels of oxidation to tissues without plaques [62].

The sentence should be reformulated.

In samples from diagnosed AD patients, the plaques and their surrounding always showed the presence of oxidized lipids. In samples from non-AD individuals showed lower levels of lipid oxidation than those from AD individuals. However, in some non-AD individuals plaques can be detected in the brain but their surrounding lipids show similar levels of oxidation to tissues without plaques [62].

Comments on the Quality of English Language

minor revision

Round 2

Reviewer 1 Report

Comments and Suggestions for Authors

The authors have addressed most of my suggestions. After some minor changes it should be publishable.

The figure legend should give a brief description of the figure. For example, “Figure 1. Mechanisms of cellular and mitochondrial iron transport. This is the title of the figure. The authors should indicate what are the upper and lower panels, and with a few sentences describe their functions/processes. Lines 111-112, ´Nevertheless, whether this pathway plays a role in brain cells such as neurons remains unclear (Figure 1)”. I think figure 1 is misplaced in this context. Please insert this figure before or after the description of the “Mechanisms of cellular and mitochondrial iron transport”.

Line 333, please define in 1-2 sentences what is ferroptosis.

Reviewer 2 Report

Comments and Suggestions for Authors

The authors accepted some of the suggested modifications and the work improved somewhat.  I did not appreciate that they did not highlight all the modifications they applied.  Some further corrections should be done.

- Line 203:  "APP has a ferric oxidase activity"  correct with ferroxidase.  it is hard to oxidise Fe(III)!!!!

- L. 378: "IR refers to the inability of target tissues to produce a normal [115]."  something is missing

- L. 383: "Wan et al. found that Fe2+-containing chloride"  I guess it is FeCl2, ferrous chloride. 

- L. 400: "Eugenio et al. found that"  Eugenio is the first name, use the family name.

- L. 465: " for treating hereditary hemochromatosis [132]" it is for treating iron overload.  Hemochromatosis is normally treated by venesection

Comments on the Quality of English Language

language is acceptable
